# The Synergistic Impact of Glycolysis, Mitochondrial OxPhos, and PEP Cycling on ATP Production in Beta Cells

**DOI:** 10.3390/ijms26041454

**Published:** 2025-02-10

**Authors:** Vladimir Grubelnik, Jan Zmazek, Marko Marhl

**Affiliations:** 1Faculty of Electrical Engineering and Computer Science, University of Maribor, Koroška cesta 46, 2000 Maribor, Slovenia; vlado.grubelnik@um.si; 2Faculty of Natural Sciences and Mathematics, University of Maribor, Koroška cesta 160, 2000 Maribor, Slovenia; jan.zmazek@um.si; 3Faculty of Medicine, University of Maribor, Taborska ulica 8, 2000 Maribor, Slovenia; 4Faculty of Education, University of Maribor, Koroška cesta 160, 2000 Maribor, Slovenia

**Keywords:** beta cell, model, ATP microdomain, anaplerosis, NADPH

## Abstract

Pancreatic beta cells regulate insulin secretion in response to glucose by generating ATP, which modulates ATP-sensitive potassium channels (K_ATP_) channel activity and Ca^2+^ dynamics. We present a model of ATP production in pancreatic beta cells, focusing on ATP dynamics within the bulk cytosol, submembrane region, and microdomains near K_ATP_ channels. ATP is generated through glycolysis, mitochondrial oxidative phosphorylation (OxPhos), and glycolytic pyruvate kinase-mediated phosphoenolpyruvate (PEP) production, supported by PEP cycling between mitochondria and the cytosol. The model examines ATP production in relation to Ca^2+^ oscillations, elucidating their interdependent dynamics. Our findings demonstrate that both mitochondrial OxPhos and PEP-mediated ATP production contribute substantially to cellular ATP levels. Specifically, glycolysis and mitochondrial OxPhos are crucial for the initial (first-phase) increase in bulk and subplasmalemmal ATP, effectively “filling up” the ATP pool in beta cells. In the second phase, coordinated cycling between OxPhos and PEP pathways enables cost-effective fine-tuning of ATP levels, with localized effects in the K_ATP_ channel microdomains. This model addresses and clarifies the recent debate regarding the mechanisms by which sufficient ATP concentrations are achieved to close K_ATP_ channels in glucose-stimulated beta cells, offering novel insights into the regulation of energy production and K_ATP_ channel activity.

## 1. Introduction

Pancreatic beta cells secrete insulin, a hormone that plays a crucial role in metabolic regulation by controlling glucose uptake, fatty acid storage, and amino acid distribution in the body. It has long been known that, after glucose stimulation, beta cells experience an increase in intracellular ATP concentration. Once ATP reaches a threshold level, it closes plasma membrane ATP-sensitive potassium channels (K_ATP_). The closure of K_ATP_ channels triggers membrane depolarization, activating voltage-gated Ca^2+^ channels. Elevated Ca^2+^ concentrations then signal the activation of cellular machinery, ultimately leading to insulin secretion. For a recent model linking key metabolic substrates—glucose, free fatty acids, and glutamine—to insulin secretion, see [1]. This model highlights the complex interplay between ATP production and anaplerotic pathways that are essential for metabolite sensing and insulin secretion in response to elevated metabolite concentrations.

While the beta cell response to high glucose is well-described and widely accepted, the exact pathways and mechanisms of ATP production remain less understood. Specifically, there is no unified explanation of how ATP is produced to close K_ATP_ channels. A straightforward possibility is that global intracellular ATP increases sufficiently to close the channels. However, experimental measurements confirm that submembrane ATP concentration rises after glucose stimulation [2,3], while the bulk ATP is only moderately changed [4]. Evidence suggests that mitochondria, particularly a subpopulation located near the plasma membrane, play a significant role in this process [5,6]. The close proximity of these mitochondria to K_ATP_ channels indicates that local ATP production in the subplasmalemmal region could suffice to close the channels without the need for a large increase in bulk ATP concentration. This localized production represents an energy-efficient mechanism, which is crucial for beta cells and aligns with the principles of cellular energy conservation.

Recently, studies have demonstrated that ATP production can occur even closer to K_ATP_ channels, in microdomains where pyruvate kinase (PK) is located [7,8]. PK catalyzes the conversion of phosphoenolpyruvate (PEP) to pyruvate, producing ATP locally in these microdomains. This localized ATP production can efficiently achieve the high concentrations required to close K_ATP_ channels. A detailed model describing this process, the MitoCat-MitoOx model, has been developed by Merrins et al. [9]. This model accounts not only for glycolytic PEP production but also for mitochondrial PEP production, an anaplerotic pathway involving the carboxylation of pyruvate to oxaloacetate (OAA) by pyruvate carboxylase (PC) followed by the conversion of OAA to PEP by mitochondrial PEP carboxykinase (PCK2), in the so-called PEP cycle [10,11,12,13]. This pathway effectively drives the tricarboxylic acid cycle (TCA) cycle in an “anticlockwise” direction, emphasizing the role of anaplerotic fluxes, as also highlighted by Grubelnik et al. [1].

A recent debate in the literature has focused on the importance of ATP production in microdomains near K_ATP_ channels versus the role of mitochondrial ATP in the subplasmalemmal region. Specifically, researchers have questioned whether PK-mediated ATP production in microdomains is essential or whether mitochondrial ATP alone suffices to close K_ATP_ channels and trigger insulin secretion [14,15,16].

In this study, we evaluate the processes and fluxes that regulate ATP concentrations in the key cellular compartments: the bulk cytosol, the subplasmalemmal region, and the microdomains near K_ATP_ channels. We integrate experimental findings and existing models to provide a comprehensive framework. Our results demonstrate that previously reported observations align logically when considered in the appropriate context. Importantly, we distinguish between the first and second phases of beta cell response.

In the first phase, the cell must “fill up” its ATP reserves, requiring significant glycolytic and mitochondrial oxidative phosphorylation (OxPhos) activity to raise ATP concentrations in both the bulk cytosol and subplasmalemmal region. In the second phase, when global ATP levels are higher, a fine-tuned interplay between mitochondrial OxPhos and PEP production (from both glycolytic and mitochondrial sources) plays a dominant role. We discuss the advantages of this fine-tuned process, emphasizing its energy efficiency, particularly during the prolonged second phase of beta cell response to elevated glucose.

This paper is organized as follows: First, we present the model, which describes the primary ATP pools and the fluxes associated with ATP production and consumption. Next, we present the results in three sections: (1) processes in the first phase, (2) dynamics during the second phase, and (3) the cyclic production of NADH and NADPH. We link NADH to mitochondrial OxPhos and NADPH to the anaplerotic “anticlockwise” TCA cycle. Notably, our model enables the separate prediction of NADH and NADPH concentrations, which are experimentally indistinguishable and measured as NAD(P)H. By summing these predicted concentrations, we validate the model against experimental data for NAD(P)H and other metabolic substrates.

## 2. Results

Our results demonstrate that in the first phase of the beta cell response to elevated glucose, glycolysis, and mitochondrial OxPhos play a crucial role in generating ATP, which needs to “fill up” the cell to establish conditions necessary for beta cell function, including insulin secretion controlled by Ca^2+^ signaling and cataplerotic/anaplerotic fluxes in the TCA cycle. The process begins with glucose-derived pyruvate being channeled into the “clockwise” TCA cycle, generating NADH and FADH_2_ as energetic molecules that fuel the electron transport chain to produce ATP. This first phase, associated with the first Ca^2+^ pulse, culminates in a significant increase in cellular ATP, particularly in the primary ATP compartments: bulk cytosolic ATP (${ATP}_{bulk}$) and ATP at the plasma membrane (${ATP}_{pm}$). This rise in ATP inhibits the “clockwise” TCA cycle by downregulating pyruvate dehydrogenase (PDH) activity, shifting the TCA cycle into the “anticlockwise” direction. This shift activates PC, redirecting pyruvate into the PEP cycle, which translocates mitochondrial ATP into the microdomains (${ATP}_{\mu d}$) near K_ATP_ channels.

The activation of the PEP cycle represents the terminal step of the first phase and the entry into the second phase of the beta cell response to glucose stimulation. The second phase is characterized by a repetitive cycling of PEP cycle activity and OxPhos phases. This phase relies on the fine regulation of ATP production across all compartments, with the most precise regulation occurring in the microdomains (${ATP}_{\mu d}$). During this phase, the cell can reduce the glycolytic flux of fresh carbon, operating in a highly energy-efficient manner to regulate insulin secretion. We provide a rough estimate of the energy cost-efficiency achieved through local ATP (${ATP}_{\mu d}$) regulation, demonstrating that the interplay with PEP cycling allows the cell to “reduce costs” while maintaining the signaling needed for effective insulin secretion.

### 2.1. Glycolytic and Mitochondrial ATP Production in the First Phase

In beta cells, stimulation with glucose triggers a biphasic response characterized by ATP production, Ca^2+^ signaling, and insulin secretion. We emphasize that the first-phase response differs dynamically from the second-phase response. Experimental studies have shown that during the first phase, ATP production precedes the Ca^2+^ response. In contrast, during the second phase, Ca^2+^ dynamics take precedence over ATP production [2]. Our model, in alignment with these experimental findings, enables detailed flux analyses and determination of the resulting concentrations. Figure 1 illustrates the interplay between ATP fluxes and Ca^2+^ dynamics, specifically considering ATP production via mitochondrial OxPhos and ATP consumption by ATPases (see the central Figure 1B).

At the onset of the first phase, ATP production is dominated by glycolysis and OxPhos, driven by the “metabolic push” initiated by glucose (Figure 1A). The ATP generated from glycolytic PEP (*PEP_Gly_*) and mitochondrial OxPhos significantly exceeds the smaller contribution from mitochondrial PEP (*PEP_m_*) production. Overall, ATP production surpasses the minimal ATP consumption by ATPases, which remain relatively inactive before the first Ca^2+^ spike. This phase is characterized by glycolysis and OxPhos “pumping” ATP into the system. During this phase, glucose undergoes glycolysis and rapid oxidation, resulting in a swift rise in ATP production that contributes to a gradual increase in bulk ATP concentration (${ATP}_{bulk}$), a more pronounced increase in subplasma membrane ATP concentration (${ATP}_{pm}$), and ultimately an ATP rise in the microdomains near the K_ATP_ channels (${ATP}_{\mu d}$). This localized ATP increase leads to the closure of K_ATP_ channels, triggering the first Ca^2+^ spike (Figure 1C). However, despite the rapid glycolytic ATP production and the OxPhos activity at the beginning of the first phase, the initial rise in ATP concentration within the microdomains surrounding the K_ATP_ channels barely reaches the threshold required to close these channels. This is due to the high diffusion of ATP away from the subplasmalemmal region, where ATP concentrations are initially low (Figure 1C).

After the first Ca^2+^ spike, ATPases are activated, primarily to pump Ca^2+^ into the endoplasmic reticulum (ER). The resulting ATP consumption causes a transient decrease in ATP concentration, which activates mitochondrial OxPhos to replenish ATP. This transition signifies the metabolic shift to the “metabolic pull” phase, where ATPase activity and Ca^2+^ dynamics drive ATP production. In this phase, ATP demand activates OxPhos, pulling pyruvate into the TCA cycle to sustain ATP generation (Figure 1B).

It is important to emphasize that following the initial closure of K_ATP_ channels and the first Ca^2+^ spike, ATP levels first transiently decrease due to the activation of ATPases but then continue to rise. The elevated ATP concentrations, combined with the gradual reduction in ATPase activity following the diminished Ca^2+^ dynamics, lead to a slowdown in OxPhos and further shift the TCA cycle toward its reverse direction. This metabolic transition marks the onset of the anaplerotic phase and the upregulation of the PEP cycle. High concentrations of *PEP_m_* drive localized ATP production, further elevating ATP levels within the microdomains surrounding the K_ATP_ channels (${ATP}_{\mu d}$). The localized ATP increase initiates the second Ca^2+^ spike, signaling the start of the second phase (Figure 1B).

Throughout the second phase, bulk ATP concentrations remain elevated, albeit with slight oscillations. Mitochondrial PEP cycling functions as a finely tuned mechanism, providing additional ATP on top of the already elevated subplasmalemmal ATP levels. This localized ATP production ensures that, due to the oscillatory nature of the process, ATP concentrations within the microdomains near the K_ATP_ channels (${ATP}_{\mu d}$) alternately reach sufficiently high levels to close the channels and trigger subsequent Ca^2+^ pulses.

### 2.2. Role of PEP Cycle and Local ATP Production in the Second Phase

As noted in the previous section, the first phase is characterized by ATP being “*filled up*” in the cell, primarily driven by a large glycolytic flux that provides a high influx of glucose-derived carbon and energy. The end products of glycolysis, PEP and pyruvate, serve as key sources of ATP: PEP is directly converted into pyruvate via PK, generating ATP locally, while pyruvate is channeled into the TCA cycle via PDH, contributing to OxPhos-driven ATP production. In contrast, the second phase is marked by a finely tuned interplay between the PEP cycle and alternating OxPhos activity. This interplay enables energy-efficient and precise regulation of K_ATP_ channel activity and Ca^2+^ dynamics, ensuring effective beta cell function. The detailed mechanisms underlying this regulatory interplay and their implications for cellular function are explored in the following section. Figure 2 illustrates the ATP production dynamics in the first and second phases. In Figure 2A, the primary processes involved in these pathways are highlighted, while Figure 2B presents an analysis of the associated flux and concentration dynamics.

The first phase of the beta cell response to high glucose corresponds to the initial Ca^2+^ pulse and encompasses the metabolic pathway “Push OxPhos”–“Pull OxPhos”–“Anaplerotic PEP Cycle”, denoted as Steps 1-2-3 in Figure 2A. Steps 1 (marked with a grey circle in Figure 2) and 2 (marked with a red circle in Figure 2) are oxidative processes characterized by “Push OxPhos” and “Pull OxPhos”. Step 1 represents the “metabolic push” of glucose directly driving OxPhos activity, whereas in Step 2, OxPhos is “pulled” by high ATPase activity following the first Ca^2+^ spike in the cell. By the end of Step 2, ATPase activity slows down, allowing OxPhos flux to exceed ATPase flux, leading to a rise in ATP levels. This increase in ATP slows down glycolysis and PDH activity, effectively reducing the “clockwise” flux of the TCA cycle. This marks the transition to Step 3 (marked with a blue circle in Figure 2), representing the “Anaplerotic PEP Cycle”, characterized by a significant pyruvate flux into the PEP cycle via PC and an increase in mitochondrial PEP flux ($J_{{PEP}_{m}}$) into the cytosol. The rise in cytosolic PEP, particularly near K_ATP_ channels, supports local ATP production via PK. This localized ATP increase ${ATP}_{\mu d}$ to the threshold value (${ATP}_{\mu d}={ATP}_{th}$), closes the K_ATP_ channels and triggers a new Ca^2+^ spike, marking the start of the second phase.

The second phase operates as a repetitive cycling process of “Pull OxPhos”–“Anaplerotic PEP Cycle” (denoted as cycling steps 2–3 in Figure 2A). In Step 2, pyruvate enters the TCA cycle in the “clockwise” direction via PDH, producing NADH and FADH_2_, which are channeled into the electron transport chain (ETC) for ATP production. This OxPhos phase is “pulled” by high ATPase activity, which sequesters Ca^2+^ into the ER and partially removes it from the cell (Figure 2B). At the beginning of Step 2, ${ATP}_{bulk}$ and ${ATP}_{pm}$ are at their maximal levels and ${ATP}_{\mu d}$ reaches the threshold value (${ATP}_{\mu d}={ATP}_{th}$), triggering the Ca^2+^ spike. The elevated Ca^2+^ concentration immediately after the spike requires increased ATPase activity to sequester Ca^2+^ into the ER and pump it out of the cell. As ATPase activity exceeds ATP production by OxPhos, ${ATP}_{bulk}$ and ${ATP}_{pm}$ begin to decline (Figure 2B). However, despite this decline, ${ATP}_{\mu d}$ remains above the threshold (${ATP}_{\mu d}>{ATP}_{th}$) throughout Step 2, supported by the high initial flux of mitochondrial PEP ($J_{{PEP}_{m}}$). Although $J_{{PEP}_{m}}$ gradually decreases during Step 2, the K_ATP_ channels remain closed, the membrane stays depolarized, and the Ca^2+^ pulse is prolonged. This is consistent with experimental data showing reduced K_ATP_ channel conductance under depolarized membrane conditions [17,18].

In Step 3, OxPhos activity surpasses ATPase activity, allowing ${ATP}_{bulk}$ and ${ATP}_{pm}$ to rise, reaching its maximal levels by the end of this phase. The increased ATP concentration inhibits PDH activity and stimulates PC activity, effectively redirecting pyruvate into the “anticlockwise” TCA cycle via PC, fueling the “Anaplerotic PEP Cycle”. This cycle redistributes ATP from mitochondria into the microdomains near K_ATP_ channels, where PK activity is high. The conversion of PEP into pyruvate via PK further elevates local ${ATP}_{\mu d}$. By the end of Step 3, ${ATP}_{\mu d}$ is sufficiently high to close the K_ATP_ channels, triggering a new Ca^2+^ pulse and marking the return to Step 2. The repetitive cycling between Step 2 (“Pull OxPhos”) and Step 3 (“Anaplerotic PEP Cycle”) defines the entire second phase of the beta cell response to high glucose stimulation, as depicted in Figure 2.

From Figure 2, we observe that PEP production is not critically important during the first phase. Although local ATP production occurs in the microdomains near K_ATP_ channels, the diffusion of ATP is driven by a steep concentration gradient due to the relatively low bulk ATP concentration. Only temporarily does the ATP concentration in the microdomain reach the threshold value (${ATP}_{\mu d}={ATP}_{th}$), initiating the first Ca^2+^ spike. However, the cell must “*fill up*” its ATP pool to a level that enables the operation of the second phase.

During the second phase, the PEP cycle plays a leading role, characterized by a “filled-up” ATP pool throughout the cell, facilitating fine-tuned local ATP production in the microdomains near K_ATP_ channels. This anaplerotic PEP cycle effectively translocates ATP from mitochondria to these microdomains, triggering Ca^2+^ pulses. The elevated Ca^2+^ concentrations subsequently activate ATPases, which reduce ATP levels. This decrease in ATP concentration, combined with high Ca^2+^ levels, activates PDH, enhancing TCA cycle flux and promoting OxPhos. These two processes—the anaplerotic PEP cycle and the OxPhos “pulled” by ATPases—operate in an alternating manner as long as there is sufficient carbon input (e.g., glucose).

The glycolytic flux during the second phase is lower than in the first phase, as energy demands are reduced. This observation is supported by Foster et al. [11], who demonstrated that the amplitude, duty cycle, and period of Ca^2+^ oscillations increase when PC activity in mitochondria (PC2), and consequently mitochondrial PEP production, is inhibited. Inhibited PEP cycling reduces PEP delivery to the subplasmalemmal region near K_ATP_ channels, which slows ATP production via PK, delays the increase in ${ATP}_{\mu d}$, and prolongs the time required to reach the threshold value for triggering the next Ca^2+^ spike (${ATP}_{\mu d}={ATP}_{th}$).

This delay implies that more time is needed to produce sufficient ATP for the subplasmalemmal ATP concentration (${ATP}_{pm}$) to reach the threshold value required to trigger a Ca^2+^ spike (${ATP}_{pm}={ATP}_{th}$). During this longer process, greater amounts of Ca^2+^ must be released and re-sequestered, resulting in increased energy fluxes in the system. To estimate the additional power required for Ca^2+^ release and re-sequestration, experimental traces of Ca^2+^ oscillations from Foster et al. [11] were analyzed.

We modeled the experimental Ca^2+^ traces using a modified Heaviside function:(1)$$H\left(t\right)=\frac{A}{2}\left(1+\;\tan h\left(S\left(\sin\left(2\pi \frac{t}{T}-\cos\pi D\right)\;\right)\;\right)\;\right)$$
where *A* is the amplitude, *T* is period, *D* is duty cycle (fraction of each oscillation spent in the electrically active state), and *S* is parameter controlling oscillation steepness. The $H\left(t\right)$ function was fitted separately for the wild-type mice (WT) and mice with β-cell PCK2 deletion of PCK2 (PCK2-βKO) (Figure 3A).

The power ($\bar{P}$) required for Ca^2+^ handling was estimated by calculating the area under the curve (*AUC*) of ATPase flux for each individual oscillation using the formula:(2)$$\bar{P}=\frac{1}{T}AUC=\frac{1}{T}\overset{T}{\underset{0}{\int }}ATPase\left(t\right)\;dt$$
where *T* is the oscillation period, and ATPase represents the ATP flux ($ATPase=-\frac{dATP}{dt}$) governed by the Ca^2+^-dependent kinetics of calcium store refilling:(3)$$ATPase\left(\left[Ca^{2+}\right]\right)=V_{SERCA}\frac{{\left[Ca^{2+}\right]}^{n}}{{\left[Ca^{2+}\right]}^{n}+K_{m}^{n}}$$
where $V_{SERCA}$ is the maximum rate of ${\mathrm{Ca}}^{2+}$ pumping, *n* = 2 (cooperativity of calcium binding) and *K_m_* = 0.5 µM (half-saturation constant) [19].

By inserting Equation (3) into Equation (2) and calculating the relative difference between the *AUC* for WT and PCK2-βKO mice, the parameter $V_{SERCA}$ cancels out. Thus, $V_{SERCA}$ can be set to an arbitrary value, and for simplicity, we set $V_{SERCA}=1$. The calculated *AUC* values were 0.78 for WT mice and 1.20 for PCK2-βKO mice (see Figure 3B). After normalizing the *AUC* values by the duration of each oscillation period ($T_{\mathrm{WT}}=2.11$ s and $T_{\mathrm{PCK}2-\beta \mathrm{KO}}=2.74$ s), the power ratio between WT and PCK2-βKO mice was found to be 1:1.18. This indicates that calcium handling in PCK2-βKO mice requires approximately 18% more power, reflecting reduced efficiency compared to WT mice.

As shown in Figure 3, the approximately 18% energy savings facilitated by the PEP cycle and localized ATP production in microdomains are critically important. While an 18% saving might seem modest, it is essential to emphasize that the second phase of the beta cell response to elevated glucose can last several tens of minutes to hours [20]. Therefore, this efficient handling of small amounts of Ca^2+^ and the corresponding localized ATP elevations in microdomains play a crucial role in maintaining normal and healthy beta cell function, ensuring effective and sustained insulin secretion.

### 2.3. Cyclic NADH and NADPH Production in the Second Phase

The second phase of the beta cell response to high glucose operates as a repetitive cycling process of “Pull OxPhos”–“Anaplerotic PEP Cycle”, as illustrated in Figure 2. The transition from Step 2 to Step 3 is characterized by the highest rates of OxPhos activity, denoted in red in Figure 2. This notation is maintained in Figure 4, where the critical features of the OxPhos phase are highlighted. During this phase, pyruvate is channeled via PDH into the “clockwise” TCA cycle, producing NADH, which is subsequently utilized for ATP production in the ETC. Conversely, the transition from Step 3 to Step 2, denoted in blue, represents the peak activity of the PEP cycle, with pyruvate fluxing in the “anticlockwise” TCA cycle direction via PC and feeding into the PEP cycle. The “anticlockwise” TCA cycle direction is associated with NADPH production through several metabolic pathways linked to glucose and, in physiological conditions, glutamine metabolism. For a recent model of this metabolism in beta cells, see [1].

One of the strengths of this model is its ability to predict the production of NADH and NADPH in stimulated beta cells. This is particularly important because experimentally, it is challenging—if not impossible in most cases—to distinguish between NADH and NADPH concentrations. Experimental data are typically represented as NAD(P)H concentrations, indicating joint measurement of NADH and NADPH. In the experiments by Merrins et al. [21], high-resolution measurements of NAD(P)H concentrations revealed an impressive “double spike” pattern in the oscillatory traces. Our model successfully explains the “double-spike” dynamics of NAD(P)H concentrations in stimulated beta cells.

To facilitate understanding of this process, Figure 4 graphically illustrates the primary metabolic pathways contributing to NADH (red) and NADPH (blue) production. Figure 4A provides a flux analysis of the main processes involved in NADH and NADPH production: NADH production by glycolysis ($J_{\mathrm{NADH},\mathrm{Gly}}$); NADH transport from the cytosol to mitochondria via the malate-aspartate shuttle ($J_{\mathrm{mal}\text{–}\mathrm{asp}\;\mathrm{shuttle}}$); the net NADPH influx into the cytosol ($J_{\mathrm{NADPH}}$), which includes NADPH production through the anaplerotic pyruvate-malate cycle and anaplerotically driven NADPH shuttles, primarily the malic enzyme (ME) and isocitrate dehydrogenase (IDH) shuttles; and the net NADPH efflux from the cytosol ($J_{\mathrm{GSH}\;\&\;\mathrm{Biosyn}}$), which accounts for NADPH usage in glutathione (GSH) synthesis and other biosynthetic processes. These fluxes vary significantly over time, resulting in oscillatory NADH and NADPH concentrations. The phase difference between these oscillations explains the experimentally observed “double-spike” patterns in the NAD(P)H traces of activated beta cells.

To better understand the temporal dynamics of the main fluxes contributing to the cytosolic concentrations of NADH and NADPH, Figure 4B presents the fluxes $J_{\mathrm{NADH},\mathrm{Gly}}$, $J_{\mathrm{mal}\text{–}\mathrm{asp}\;\mathrm{shuttle}}$, $J_{\mathrm{NADPH}}$, and $J_{\mathrm{GSH}\;\&\;\mathrm{Biosyn}}$ in relation to the primary glycolytic input flux and the TCA cycle fluxes. During the maximum OxPhos phase, the glycolytic flux is high, and the resulting pyruvate is primarily directed via PDH into the oxidative “clockwise” TCA cycle. Glycolytic NADH production is elevated, and the NADH is shuttled into mitochondria through the malate-aspartate shuttle.

As NADH is converted to ATP and cytosolic ATP levels rise, glycolysis is inhibited. Additionally, elevated ATP levels inhibit PDH and activate PC, progressively shifting the TCA cycle toward the “anticlockwise” direction, including the PEP cycle. This anaplerotic phase generates NADPH through the pyruvate-malate cycle and facilitates NADPH transport from mitochondria to the cytosol via the ME and IDH shuttles. Concurrently, NADPH is utilized for antioxidative processes, such as GSH synthesis and other biosynthetic pathways.

## 3. Discussion

The presented model emphasizes the importance of separately considering the first and second phases of the beta cell response to elevated glucose concentrations. In the first phase, high glucose drives ATP production through a “metabolic push”, with glycolysis and mitochondrial OxPhos playing crucial roles. The initial elevation of ATP induced by this metabolic push involves substantial energy investment. This aligns with experimental measurements of oxygen consumption in beta cells, which show that oxidation and OxPhos are highly active during the early response to glucose stimulation [2,22].

Moreover, the model is consistent with experimental findings that highlight the significance of specific mitochondrial distribution within the cell, particularly the subpopulation located beneath the plasma membrane [5,6]. As ATP elevation occurs primarily in the subplasmalemmal region without a substantial increase in bulk ATP levels [2,3,4], our results underscore the critical role of this subplasmalemmal mitochondrial subpopulation in ATP production.

This understanding is particularly relevant for studying the pathogenesis of type 2 diabetes mellitus (T2DM), where alterations in the first-phase response to glucose stimulation in beta cells have been observed [20,23]. Additionally, the consequences of mitochondrial dysfunction in alpha and beta cells have been extensively studied in relation to T2DM [24,25].

As demonstrated by our model and supported by experimental data [2], ATP elevation in the first phase is followed by a slower and less pronounced rise in calcium. The model attributes this phenomenon to the substantial energy demand necessary to initiate the system. Following the rise in Ca^2+^ during the first phase, elevated Ca^2+^ concentrations activate ATPases to sequester Ca^2+^, primarily into the ER. This increased ATPase activity consumes significant amounts of ATP, effectively “pulling” OxPhos to sustain ATP production via the TCA cycle. This ATPase-driven “pull” on oxidative processes continues throughout the second phase.

The second phase begins after the ATP pool in the cell is “filled up”, providing the foundation for fine-tuned ATP production in microdomains near K_ATP_ channels. In this phase, ATP is locally produced by converting PEP into pyruvate via PK. The model’s predictions align with experimental findings showing that PK is located close to the plasma membrane, where K_ATP_ channels are situated [8,9,11]. Local ATP production in these microdomains is also energetically advantageous. According to our model calculations, energy savings of approximately 18% were achieved when comparing wild-type beta cells to PCK2-βKO mice, based on calcium oscillation data from Foster et al. [11]. This energy conservation is particularly important during the second phase, which is long-lasting and essential for sustained insulin secretion, especially under conditions of high metabolic demand. Although this 18% is a rough estimate, it appears to be realistic given the prolonged nature of the second phase and the associated energy requirements.

Calcium transport across compartments and ATPase activity are inherently energy-intensive. For instance, recent studies on adipocytes demonstrated that a so-called “Ca^2+^ futile cycle” induced by GIPR activation significantly increased energy consumption, leading to weight loss of approximately 35% [26]. Thus, localized ATP production near K_ATP_ channels is critical for energy efficiency. Moreover, recent research highlights the importance of highly localized compartments in glucose-stimulated beta-cell processes. For example, human islets exhibit primary cilia observed by electron microscopy, which may act as vital glucose-sensing organelles [27].

The second phase, a long-lasting and energy-efficient phase, is critically characterized by anaplerotic pathways. In addition to the PEP cycle highlighted in our model, other anaplerotic pathways play significant roles in enabling beta cells to sense metabolite abundance and produce substantial amounts of insulin in response to food intake under physiological conditions [1]. For example, citrate—one of the key anaplerotic metabolites derived from the TCA cycle—has been experimentally shown to oscillate in an antiphase with Ca^2+^ [28]. This observation aligns with our model’s predictions, which show NADPH—another critical metabolite produced via the “anticlockwise” TCA cycle—exhibiting similar oscillatory behavior.

Additionally, our model’s depiction of glycolytic flux aligns well with experimentally measured glycolytic intermediates, such as fructose-1,6-bisphosphate, which oscillates in phase with cytosolic Ca^2+^ concentrations [21]. The high glycolytic flux, coupled with elevated OxPhos activity (as shown by the in-phase dynamics in Figure 2B), underscores the critical role of mitochondrial ATP production. It is important to emphasize that our model does not contradict the “traditional” view of beta cell function—referred to by Merrins et al. [9] as the “canonical” model—in which OxPhos raises the ATP/ADP ratio to trigger K_ATP_ channel closure, Ca^2+^ influx, and insulin release. This “canonical” model has been the subject of recent discussions about its validity [14,15,16]. Rather, our model demonstrates that the “canonical” model represents only part of the complex metabolic response of beta cells to high glucose. Specifically, while the “canonical” model elucidates mitochondrial oxidative ATP production during the “push-OxPhos” first phase, our model extends this understanding by incorporating the “pull-OxPhos” phase, in which ATPases drive NADH production and subsequent ATP synthesis via the ETC.

Our model provides a more comprehensive perspective on the intricate metabolic processes in beta cells following glucose stimulation. This approach integrates previous models and experimental findings, offering a unified explanation for observations obtained under varying conditions and at different time points during beta cell responses. For example, some experimental results focus on the early response (first phase), while others pertain to the prolonged response in the second phase.

Importantly, our model captures the dynamics of key fluxes that regulate the concentrations of Ca^2+^, ATP, PEP, and other metabolites. These temporal dynamics can be challenging to intuit because substrate concentrations arise from a balance between influx and efflux processes. Moreover, the model highlights the importance of compartmentalized metabolite production. For instance, high NADH concentrations in the cytosol during OxPhos activity might be misinterpreted as being directly caused by mitochondrial NADH production. However, mitochondrial NADH is confined to mitochondria, as NADH cannot cross the inner mitochondrial membrane. Instead, cytosolic NADH increases are primarily driven by glycolysis.

The strength of the model is also its ability to predict separate concentrations of NADH and NADPH in the cytosol. Experimentally, these metabolites are typically measured as their combined signal, NAD(P)H, without the ability to distinguish between the two. By predicting NADH and NADPH separately and summing them, the model achieves a direct match with experimental NAD(P)H data from Merrins et al. [21]. This is a significant result that provides insights and directions for future experimental studies. As experimental techniques advance, the model’s predictions can be further validated, demonstrating how theoretical approaches can inspire new experimental research. This synergy between theoretical and experimental studies fosters scientific development and opens pathways for new discoveries.

For future studies, separating the measurements of NADPH and NADH and particularly analyzing the role of NADPH in beta cells remains one of the key challenges. NADPH plays multiple crucial roles in beta cells, including its involvement in the triggering phase of glucose-stimulated insulin secretion. It has been shown that H_2_O_2_ produced from NADPH via the NADPH oxidase isoform 4 (NOX4) is essential for insulin secretion [29]. On the other hand, at higher rates of OxPhos and concomitant increased ROS production, NADPH is required for glutathione (GSH) synthesis, which is crucial for the antioxidative protection of beta cells. This highlights the dual role of NADPH in both H_2_O_2_ production and its neutralization [30]. Moreover, NADPH, produced as a key TCA-derived product of anaplerosis, also plays a critical role in the biosynthesis of fatty acids and other biomolecules. All these pathways were recently modeled under stationary conditions [1] and should be further extended to incorporate the temporal—oscillatory—dynamics of the metabolites studied in this paper.

In the future, it would also be valuable to apply the predictions of the model presented here to pathophysiological conditions, particularly T2DM. We emphasize the role of hypoxic conditions related to T2DM, as elevated glucose levels are already known to induce hypoxia in pancreatic beta cells, leading to a decrease in oxygen partial pressure (pO_2_) due to increased glucose oxidation [31,32]. Additionally, hypoxia is associated with the accumulation of succinate, primarily due to succinate dehydrogenase (SDH) deficiency [33]. Understanding the role of succinate accumulation and the activation of HIF1α and its downstream target genes under hypoxic conditions [34] could also be extended to other cellular systems. Recent studies have drawn parallels between stimulated beta cells and cancer cells [35]. Further exploration of these parallels and the broader application of beta cell research findings may enhance our understanding of other physiological systems and contribute to advancements in both scientific research and clinical treatments.

## 4. Materials and Methods

The model of glucose-stimulated beta cells encompasses the processes of glycolysis, mitochondrial OxPhos, and anaplerotic processes, with an emphasis on the PEP cycle. After glycolysis, which converts glucose to pyruvate, the fate of pyruvate diverges into two primary pathways: entering the oxidative pathway via PDH or participating in the anaplerotic pathway via PC into the PEP cycle (Figure 5A).

In the model, we focus on ATP production at different scales: the global concentration in the cell (${ATP}_{bulk}$), at the plasma membrane (${ATP}_{pm}$), and in the microdomains close to K_ATP_ channels (${ATP}_{\mu d}$). Since ATP can diffuse through the cytosol, these three regions are conceptualized as interconnected “leaky” compartments, where the fluxes of ATP into, out of, and between the compartments play a critical role in reaching the threshold ATP concentration (${ATP}_{th}$) required to activate the K_ATP_ channels (Figure 5B).

Glucose entering the cell serves as a substrate for glycolysis, where PEP is produced. In the final step of glycolysis, PEP is converted into pyruvate by PK. This conversion, which occurs near K_ATP_ channels where PK is located, contributes to the localized rise in ATP concentration within the microdomains (${ATP}_{\mu d}$) [8]. Based on the molecular weights of pyruvate (approximately 88 g/mol), PEP (around 168 g/mol), and ATP (about 507 g/mol) [36], we estimate that pyruvate diffuses significantly more efficiently than PEP or ATP, with approximate relative diffusion factors of 20% and 60%, respectively, compared to pyruvate. As a result, pyruvate, serving as a substrate for the TCA cycle, can readily diffuse to mitochondria located throughout the cell, particularly to mitochondria near the plasma membrane [5,6] and in proximity to ATPases, where it supports ATP production for their activity.

In addition to its role in the OxPhos pathway through the “clockwise” direction of the TCA cycle, pyruvate also enters the TCA cycle in its “anticlockwise” direction, producing PEP via PC in the PEP cycle. As illustrated in Figure 5, the red-labeled OxPhos phase and the blue-marked PEP cycle alternate in their activities, resulting in oscillatory patterns of ATP production across different cellular compartments (${ATP}_{bulk}$, ${ATP}_{pm}$, and ${ATP}_{\mu d}$).

In this study, we take a complementary phenomenological approach, reconstructing oscillatory metabolite dynamics directly from experimental data. Rather than deriving behavior from first-principles equations, we infer unknown metabolite curves based on observed oscillation maxima, phase shifts, and periodicities. This method ensures consistency with experimental trends while avoiding overparameterization, a challenge often encountered in mechanistic models. Compared to our previous mechanistic model, this approach captures metabolic oscillations with fewer assumptions, emphasizing empirical fidelity over biochemical detail. By refining the phase relationships and amplitudes of key metabolites, we generate a representation of beta cell metabolism that aligns closely with experimental observations.

The ATP dynamics in different compartments are modeled based on previously published experimental data and established models describing ATP dynamics. ATP concentrations have been measured in the bulk cytosol (${ATP}_{bulk}$) and the subplasmalemmal region (${ATP}_{pm}$), providing a foundation for this model [2,3,4,5,37]. The metabolic fate of pyruvate, ATP production via the electron transport chain (ETC), and the roles of TCA-derived NADH and FADH_2_ have also been well described and modeled in previous studies (for a recent model, see [1]).

In addition to the oxidative fate of pyruvate-derived products, Grubelnik et al. [1] also modeled the metabolism of free fatty acids and amino acids, as well as anaplerotic pathways, including the conversion of pyruvate to OAA via PC, the PEP cycle, and NADPH production. Our qualitative model integrates and synthesizes this knowledge, providing a holistic view of the metabolic processes governing beta cell function.

A major advantage of our methodological approach is the ability to build a model with the capacity to integrate diverse experimental findings into a unified framework, allowing for the simultaneous consideration of metabolic processes occurring in beta cells. Rather than solely building a model that focuses on fitting ATP traces, our approach emphasizes an integrative framework that incorporates a wide range of experimentally measured parameters, including Ca^2+^ dynamics—one of the most frequently studied signaling molecules—as well as PEP levels, oxygen consumption rate (OCR), and NAD(P)H oscillations [8,11,21,22,28,38,39,40]. These experimental measurements are systematically integrated into the Results section, where they are analyzed in conjunction with our model predictions.

This modeling approach enables the simultaneous incorporation of time delays between the oscillatory patterns of different metabolites—including Ca^2+^, ATP, PEP, NADH, and NADPH. By contextualizing these interactions, our model provides deeper insights into beta cell metabolism beyond simple concentration changes. Additionally, it qualitatively evaluates the corresponding fluxes, offering a more dynamic perspective on metabolic regulation. The fundamental framework for these flux analyses is derived from the recent work of Grubelnik et al. [1].

In the Results section, we present both the most relevant temporal dynamics of metabolites and the corresponding fluxes that contribute to their regulation. This approach ensures that our model not only aligns with experimental observations but also provides new perspectives on the intricate metabolic interactions that drive beta cell responses to glucose stimulation.

## Figures and Tables

**Figure 1 ijms-26-01454-f001:**
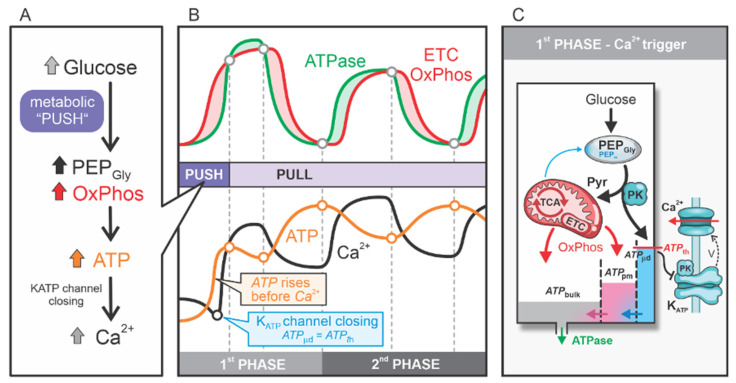
ATP “filling up” during the first phase in a glucose-stimulated beta cell, driven by the “metabolic push” of glucose through high glycolytic flux and elevated OxPhos activity. The subsequent “pull phase” occurs when ATPase activity “pulls” OxPhos to meet the ATP demand. In the second phase, bulk ATP concentrations remain elevated but exhibit slight oscillations due to the alternating activity of OxPhos and PEP cycle activation. (**A**) Glucose-driven “push” on glycolysis and OxPhos. (**B**) ATP and Ca^2+^ dynamics during the 1st and 2nd phases. (**C**) ATP fluxes into and out of the three pools: ${ATP}_{bulk}$, ${ATP}_{pm}$, and ${ATP}_{\mu d}$.

**Figure 2 ijms-26-01454-f002:**
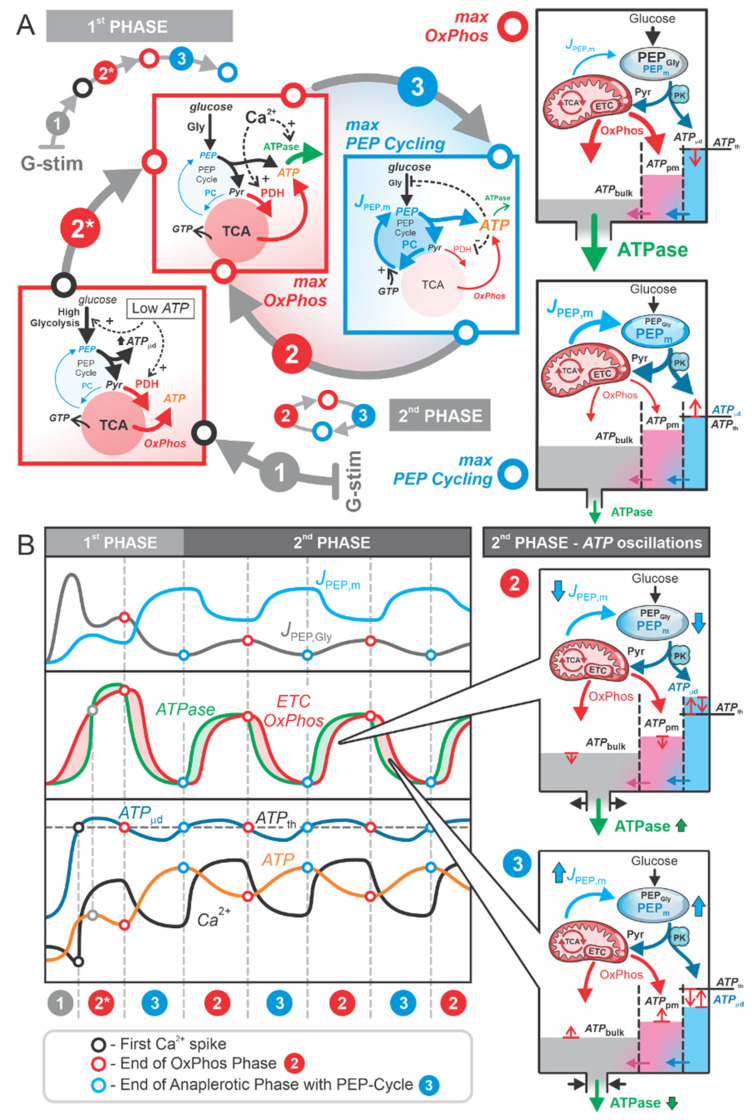
ATP production in a glucose-stimulated beta cell. (**A**) The first phase consists of three steps: (1) “Push OxPhos”, (2) “Pull OxPhos”, and (3) “Catabolic PEP Cycle”. The second phase is characterized by a cycling process alternating between steps 2 and 3. The OxPhos step is represented by a solid red circle, while the PEP cycle is denoted by a solid blue circle. The maximum OxPhos state is marked with a hollow red circle, and the maximum PEP cycling state is marked with a hollow blue circle. Both transitions 2 and 2* denote the “Pull OxPhos” step; however, since this step starts differently in the first phase, 2* is used to distinguish it from the periodically repeated “Pull OxPhos” steps in the second phase. (**B**) Fluxes contributing to ATP production: $J_{{PEP}_{m}}$ (blue), $J_{{PEP}_{Gly}}$ (black), ATPase flux (green), and ETC OxPhos flux (red). Also shown are the concentrations of subplasma cytosolic ATP (${ATP}_{pm}\approx {ATP}_{bulk}$), the ATP threshold (${ATP}_{th}$), and Ca^2+^.

**Figure 3 ijms-26-01454-f003:**
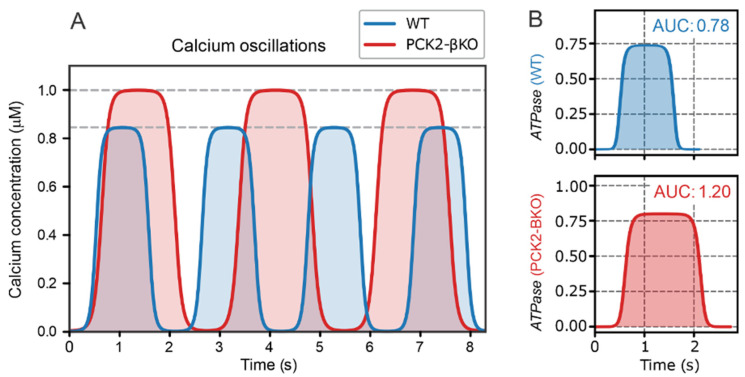
Estimation of the translocated Ca^2+^ amount, used as a proxy for energy expenditure, in PCK2-βKO and control (WT) mice as studied in the experiments by Foster et al. [11]. (**A**) Fitted Ca^2+^ oscillation traces for WT and PCK2-BKO mice using the modified Heaviside function H(t). For WT, the parameters used were A = 0.847, D = 0.496, P = 2.111, and S = 3, and for PCK2-βKO, the parameters used were A = 1, D = 0.532, P = 2.740, and S = 3, highlighting differences in amplitude, period, and duty cycle. (**B**) Energy expenditure of calcium pumps during oscillations, calculated as the area under the curve (AUC) for the refilling kinetics $ATPase\left(\left[Ca^{2+}\right]\right)$. WT mice exhibit lower energy expenditure compared to PCK2-βKO mice, with a normalized power ratio of 1:1.18, indicating a ~18% increase in energy demand for PCK2-βKO mice.

**Figure 4 ijms-26-01454-f004:**
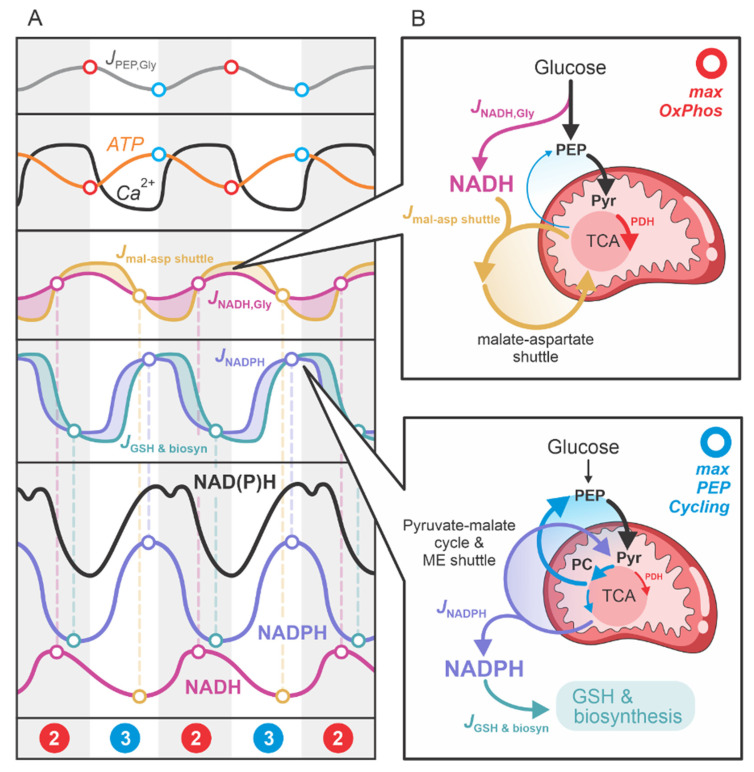
Graphical representation of the metabolic pathways contributing to NADH (red) and NADPH (blue) production. (**A**) Flux analysis of the main processes responsible for NADH and NADPH production: $J_{\mathrm{NADH},\mathrm{Gly}}$, $J_{\mathrm{mal}-\mathrm{asp}\;\mathrm{shuttle}}$, $J_{\mathrm{NADPH}}$, and $J_{\mathrm{GSH}\;\&\;\mathrm{Biosyn}}$. (**B**) Illustration of the main fluxes associated with the “clockwise direction” of the TCA cycle during the maximum OxPhos phase and the “anticlockwise direction” of the TCA cycle during the PEP cycle phase.

**Figure 5 ijms-26-01454-f005:**
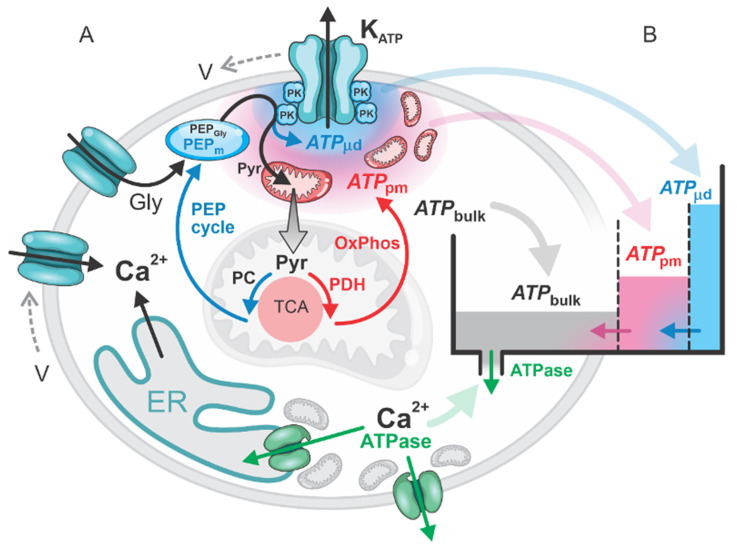
Schematic representation of the key energetic processes in a glucose-stimulated beta cell. (**A**) ATP is produced via glycolysis (Gly) and mitochondrial oxidative phosphorylation (OxPhos) and translocated by the PEP cycle, with its primary utilization by ATPases. (**B**) The model considers three main pools of ATP: the global cytosolic concentration (${ATP}_{bulk}$), the subplasmalemmal concentration near the plasma membrane (${ATP}_{pm}$), and the localized concentration in the microdomains near K_ATP_ channels (${ATP}_{\mu d}$).

## Data Availability

Data is contained within the article.

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
