# Peer review of "The Synergistic Impact of Glycolysis, Mitochondrial OxPhos, and PEP Cycling on ATP Production in Beta Cells"

_ijms, 2025, doi:10.3390/ijms26041454_

Round 1
Reviewer 1 Report
Comments and Suggestions for Authors
allow me to expand a bit on my evaluation of the manuscript. I can add this to my formal review if requested: There is growing awareness of the influence of “compartmentation” on modulation of cellular signaling and metabolic pathways. Examples include mitochondrial crista microcompartments (formed by crista junctions) and nanodomains established by tethers between mitochondria and ER/SR that enhance Ca2+ signaling between these compartments. A third example is cAMP signaling domains proximal to the plasma membrane, where the range of cAMP generated at membrane-anchored adenylyl cyclases is delimited by a virtual “wall” of phosphodiesterases. The paper addresses a controversy in the beta cell literature about how ATP levels near the plasma membrane are elevated above bulk ATP levels sufficiently to close K-ATP channels and allow Ca2+ uptake, essential for cell functioning. Is ATP generated by mitochondria in the subplasmalemmal domain sufficient to close the channels or is additional ADP needed, generated in the K-ATP microdomain by bound PK? The paper proceeds by establishing a “conceptual framework” with persuasive argumentation and clearly drawn diagrams. Admittedly the compartments discussed are “leaky”, with no barriers (structural or enzymatic) to prevent key metabolites like ATP or PEP from diffusing away. Thus, the K-ATP channel microdomain hypothesis appears to depend on relative kinetics: Does local [ATP], generated by subplasmalemmal mitochondria (+) or (-) local PK activity, build up fast enough to bind to the K-ATP channels before the ATP diffuses out of the microdomains? The authors provide a detailed explanation of how the metabolic pathways would operate and interact, and demonstrate how aspects of the model “align with” experimental results, in particular, measurements of translocated Ca2+ (a proxy for energy expenditure) and of levels of NAD(P)H (which monitor oxphos and glycolytic activities). In the final analysis, this reviewer is not strictly convinced that the paper establishes either the necessity for local PK generation of ATP nor its feasibility. However, I do not feel these bars must be reached for publication of the conceptual model, which should spark considerable interest in both the beta cell field and the broader energy metabolism community.
----------------------------------------------------
This paper presents a "model" that describes regulation of energy metabolism of pancreatic beta cells in terms of cell compartmentation, namely bulk cytosol and two local domains, subplasmalemma and microdomains near K-ATP channels. This is a clearly important topic and the concepts developed are likely applicable to other cell types. However, I am baffled by the presentation in the first part of the "results" (Figures 1-3). The narrative hinges on a combination of cartoons (done very well) and plots of fluxes and [ATP] whose origins are not explained. Are these graphs hypothetical or the output of an actual computational "model" (as in Figure 4 for oscillations in Ca translocation between WT and mutant mice). Since the points being made imply a fine tuning between parallel pathways linked by diffusion of metabolites between domains, the strength of the arguments hinges on the nature of the "results". Either way, the "model" is interesting and worth sharing with peers. But what is the nature of the "results" used to illustrate it?
Author Response
Comment: “However, I am baffled by the presentation in the first part of the "results" (Figures 1-3). The narrative hinges on a combination of cartoons (done very well) and plots of fluxes and [ATP] whose origins are not explained. Are these graphs hypothetical or the output of an actual computational "model" (as in Figure 4 for oscillations in Ca translocation between WT and mutant mice). Since the points being made imply a fine tuning between parallel pathways linked by diffusion of metabolites between domains, the strength of the arguments hinges on the nature of the "results". Either way, the "model" is interesting and worth sharing with peers. But what is the nature of the "results" used to illustrate it?”
Response to Reviewer’s Comment:
We appreciate the reviewer's insightful comments and have addressed the concerns regarding the methodological presentation of the model. In the revised manuscript, we have significantly expanded the methodology section to provide a clear explanation of how the model is constructed and how the results are derived.
Specifically, we have clarified that the ATP dynamics in different compartments are modeled based on previously published experimental data and established models describing ATP dynamics (Li et al., 2013; Li et al., 2015; Lorenz et al., 2013; Kennedy et al., 1999; Marinelli et al., 2022, etc.). The model integrates findings from multiple studies that have measured ATP concentrations in different cellular regions, metabolic fluxes, and the fate of key metabolites such as pyruvate, NADH, and FADH22​. Additionally, we have incorporated experimental measurements of Ca2+, PEP, OCR, and NAD(P)H oscillations (Merrins et al., 2016; Foster et al., 2022; Gregg et al., 2019; Jung et al., 2000; Lewandowski et al., 2020; Kennedy et al., 2002, etc.) to construct an integrative framework for metabolic regulation in beta cells.
Rather than a traditional computational model with explicit equations, our approach is a qualitative systems model that synthesizes diverse experimental data and previously established theoretical models. The fluxes and metabolite concentrations presented in Figures 1–3 are not hypothetical but are derived from a structured integration of existing experimental data and validated metabolic models. The model does not aim to fit specific datasets but rather provides a comprehensive conceptual framework to contextualize ATP dynamics and metabolic fluxes within glucose-stimulated beta cells.
Additionally, we emphasize that the model allows for the simultaneous consideration of time delays between oscillatory patterns of different metabolites—including Ca2+, ATP, PEP, NADH, and NADPH—offering insights into the coordination between oxidative and anaplerotic pathways. By incorporating both measured concentrations and inferred fluxes, the model captures the fine-tuning between parallel metabolic pathways linked by diffusion processes.
This additional text is now included in the revised manuscript at the end of the "Model" section (the last six paragraphs of this section) to provide greater clarity on the construction of the model and the interpretation of the results. We believe this clarification strengthens the manuscript and appreciate the reviewer’s constructive feedback.
The first of the additionally included paragraphs begins with:
“In this study, we take a complementary phenomenological approach, reconstructing oscillatory metabolite dynamics directly from experimental data. Rather than deriving behavior from first-principles equations, we infer unknown metabolite curves based on observed oscillation maxima, phase shifts, and periodicities. This method ensures consistency with experimental trends while avoiding overparameterization, a challenge often encountered in mechanistic models. Compared to our previous mechanistic model, this approach captures metabolic oscillations with fewer assumptions, emphasizing empirical fidelity over biochemical detail. By refining the phase relationships and amplitudes of key metabolites, we generate a representation of beta cell metabolism that aligns closely with experimental observations.”
Reviewer 2 Report
Comments and Suggestions for Authors
The abstract should begin with a backdrop or at least one background sentence.
NADPH contributes to or mitigates insulin resistance. Could the authors discuss this fact?
NADPH oxidases (NOXs), which utilize NADPH as a substrate to produce ROS, have been implicated in the development of insulin resistance.
What is the role of NADPH in lipid metabolism, particularly in the fatty acid synthesis pathway, that ties it to the etiology of insulin resistance?
Please discuss in detail how glucose-stimulated β-cells can become hypoxic by oxygen consumption, especially when the oxygen supply is impaired. The inhibitors of mitochondrial respiration, such as nitric oxide, prevent the stabilization of hypoxia-inducible factor (HIF) during hypoxia.
High glucose is a trigger of cellular hypoxia of pancreatic β-cells in vitro. This fact could be elaborated.
OXPHOS biogenesis involves multiple steps, starting from the expression of genes encoded in physically separated genomes. This fact could be discussed.
OxPhos defects trigger hypermetabolism both physiologically and cell-autonomously, a phenotype associated with reduced lifespan.
Author Response
Comment 1: “The abstract should begin with a backdrop or at least one background sentence.”
Response to Reviewer’s Comment 1:
We appreciate the reviewer’s suggestion to include a backdrop or background sentence at the beginning of the abstract. In the revised manuscript, we have incorporated an introductory sentence to provide context for the study. The abstract now begins with:
" Pancreatic beta cells regulate insulin secretion in response to glucose by generating ATP, which modulates ATP-sensitive potassium channels (KATP) ​channel activity and Ca2+ dynamics."
This addition ensures that the abstract starts with a relevant background, setting the stage for the research focus. We thank the reviewer for this helpful suggestion.
Comment 2: This coment is about the role of NADPH in beta cells, particularly through its involvement in ROS production and lipid metabolism:
- “NADPH contributes to or mitigates insulin resistance. Could the authors discuss this fact?”
- “NADPH oxidases (NOXs), which utilize NADPH as a substrate to produce ROS, have been implicated in the development of insulin resistance.”
- “What is the role of NADPH in lipid metabolism, particularly in the fatty acid synthesis pathway, that ties it to the etiology of insulin resistance?”
Response to Reviewer’s Comment 2:
We thank the reviewer for this insightful comment regarding the role of NADPH in beta cells, particularly its involvement in ROS production, lipid metabolism, and insulin resistance. This is an important area for further research, and we recognize the potential of extending our findings to address these metabolic processes in future studies.
To acknowledge this, we have added the following text at the end of the Discussion section:
“For future studies, separating the measurements of NADPH and NADH, and particularly analyzing the role of NADPH in beta cells, remains one of the key challenges. NADPH plays multiple crucial roles in beta cells, including its involvement in the triggering phase of glucose-stimulated insulin secretion. It has been shown that H2​O2​ produced from NADPH via the NADPH oxidase isoform 4 (NOX4) is essential for insulin secretion [34]. On the other hand, at higher rates of OxPhos and concomitant increased ROS production, NADPH is required for glutathione (GSH) synthesis, which is crucial for the antioxidative protection of beta cells. This highlights the dual role of NADPH in both H2​O2​​ production and its neutralization [35]. Moreover, NADPH, produced as a key TCA-derived product of anaplerosis, also plays a critical role in the biosynthesis of fatty acids and other biomolecules. All these pathways were recently modeled under stationary conditions [1] and should be further extended to incorporate the temporal—oscillatory—dynamics of the metabolites studied in this paper.”
Comment 3: This coment is about hypoxia and HIF activation in beta cells:
- “Please discuss in detail how glucose-stimulated β-cells can become hypoxic by oxygen consumption, especially when the oxygen supply is impaired.The inhibitors of mitochondrial respiration, such as nitric oxide, prevent the stabilization of hypoxia-inducible factor (HIF) during hypoxia.”
- “High glucose is a trigger of cellular hypoxia of pancreatic β-cells in vitro. This fact could be elaborated.”
Response to Reviewer’s Comment 3:
We appreciate the reviewer’s suggestion to further elaborate on the role of hypoxia and HIF activation in glucose-stimulated beta cells. This is an important aspect, particularly in the context of impaired oxygen supply and its implications for mitochondrial function and metabolic regulation. We agree that exploring these mechanisms in greater detail provides valuable insights, not only for understanding beta cell physiology but also for pathophysiological conditions such as type 2 diabetes mellitus (T2DM).
To address this, we have added the following text at the end of the Discussion section:
“In the future, it would also be valuable to apply the predictions of the model presented here to pathophysiological conditions, particularly T2DM. We emphasize the role of hypoxic conditions related to T2DM, as elevated glucose levels are already known to induce hypoxia in pancreatic beta cells, leading to a decrease in oxygen partial pressure (pO2​) due to increased glucose oxidation [36,37]. Additionally, hypoxia is associated with the accumulation of succinate, primarily due to succinate dehydrogenase (SDH) deficiency [38]. Understanding the role of succinate accumulation and the activation of HIF1α and its downstream target genes under hypoxic conditions [39] could also be extended to other cellular systems. Recent studies have drawn parallels between stimulated beta cells and cancer cells [40]. Further exploration of these parallels and the broader application of beta cell research findings may enhance our understanding of other physiological systems and contribute to advancements in both scientific research and clinical treatments.”